

# Estimating the incidence and diagnosed proportion of HIV infections in Japan: a statistical modeling study

Hiroshi Nishiura

Graduate School of Medicine, Hokkaido University, Sapporo, Japan
CREST, Japan Science and Technology Agency, Saitama, Japan

## ABSTRACT

**Background**. Epidemiological surveillance of HIV infection in Japan involves two technical problems for directly applying a classical backcalculation method, i.e., (i) all AIDS cases are not counted over time and (ii) people diagnosed with HIV have received antiretroviral therapy, extending the incubation period. The present study aimed to address these issues and estimate the HIV incidence and the proportion of diagnosed HIV infections, using a simple statistical model.

**Methods**. From among Japanese nationals, yearly incidence data of HIV diagnoses and patients with AIDS who had not previously been diagnosed as HIV positive, from 1985 to 2017, were analyzed. Using the McKendrick partial differential equation, general convolution-like equations were derived, allowing estimation of the HIV incidence and the time-dependent rate of diagnosis. A likelihood-based approach was used to obtain parameter estimates.

**Results**. Assuming that the median incubation period was 10.0 years, the cumulative number of HIV infections was estimated to be 29,613 (95% confidence interval (CI): 29,059, 30,167) by the end of 2017, and the proportion of diagnosed HIV infections was estimated at 80.3% (95% CI [78.7%–82.0%]). Allowing the median incubation period to range from 7.5 to 12.3 years, the estimate of the proportion diagnosed can vary from 77% to 84%.

**Discussion**. The proportion of diagnosed HIV infections appears to have not yet reached 90% among Japanese nationals. Compared with the peak incidence from 2005–2008, new HIV infections have clearly been in a declining trend; however, there are still more than 1,000 new HIV infections per year in Japan. To increase the diagnosed proportion of HIV infections, it is critical to identify people who have difficulty accessing consultation, testing, and care, and to explore heterogeneous patterns of infection.

## INTRODUCTION

Following an infection with human immunodeficiency virus (HIV), development of acquired immunodeficiency syndrome (AIDS) takes about 10 years (*Muñoz, Sabin & Phillips, 1997*). The long incubation period makes it difficult to directly observe

Corresponding author
Hiroshi Nishiura,
nishiurah@med.hokudai.ac.jp

the incidence and prevalence of HIV infections over time. To offer insights into the epidemiology of HIV-infected and -incubating individuals over time, and to evaluate public health control programs, various statistical modeling approaches have been proposed to date (*Brookmeyer & Gail, 1994*; *Donnelly & Cox, 2001*; *Jewell, Dietz & Farewell, 1992*). Of these, a backcalculation method using a simple integral equation to model AIDS incidence as arising from the HIV incidence convoluted with the independently and identically distributed incubation period allows estimation of the HIV incidence based on epidemiological surveillance data (*Brookmeyer & Gail, 1986*; *Gail & Brookmeyer, 1988*). Assuming that the reported number of AIDS cases certainly and accurately captures the actual number of AIDS incidence in industrialized countries, the backcalculation method greatly improves our understanding of the epidemiology of HIV infection, attributing the observed AIDS data to HIV infection events as a function of time.

Understanding the transmission dynamics of HIV using such statistical models is in line with the concept of treatment cascade, introduced by the Joint United Nations Programme on HIV/AIDS (UNAIDS). The so-called care cascade aims to identify and fill gaps in the continuum of services for testing, care, and effective treatment of HIV (*UNAIDS, 2014*). In relation to this, the UNAIDS report has led to the global initiative "90–90-90" by 2020 that sets out goals in care cascades to achieve the following: 90% of people living with HIV know their HIV status, 90% of people diagnosed with HIV have access to antiretroviral therapy (ART), and 90% of people receiving ART have suppressed viral loads (*UNAIDS, 2014*). UNAIDS even aims to achieve 95-95-95 at a global level by the year 2030, contributing to successfully controlling HIV and AIDS, as supported by the so-called test-and-treat strategy (*Granich et al., 2009*; *Granich et al., 2017*). To quantify the situation of each country, monitoring diagnosed individuals is essential; moreover, estimation of the diagnosed proportion of HIV infections must be supported by firm scientific methods, to estimate the first part of the three 90-90-90 targets, i.e., 90% of HIV-infected people know their HIV status. In Japan, an analysis of blood donors took place in 2017, in which it was estimated that 85.6% of HIV-infected individuals, regardless of nationality, were diagnosed (*Iwamoto et al., 2017*). Nevertheless, it is known that the analysis of voluntary blood donation data is prone to sampling bias of donors owing to the tendency of people with high risk to repeatedly undertake anonymous laboratory testing through the practice of blood donation (*Kihara, Imai & Shimizu, 2000*), and moreover, an exclusion of repeaters can result in underestimation of the prevalence, resulting in overestimation of the fraction diagnosed, even though such screening of a large number of people is very costly. Considering the need to achieve continued monitoring of the diagnosed proportion of HIV-infected individuals, development of a reasonable yet scientifically rigorous method based on other datasets would be beneficial, especially using epidemiological surveillance data (*Hsieh et al., 2012*; *Cuadros & Abu-Raddad, 2016*; *Hsieh & Lin, 2016*; *Mumtaz et al., 2018*).

Despite the clear need for epidemiological estimation of the number of undiagnosed HIV infections, the surveillance data in Japan possesses two technical problems. First, while the definition of AIDS has remained nearly unchanged over time, reporting AIDS cases that were previously diagnosed as HIV-infected cases has never been mandated

(*Nishiura, 2007*). This makes it impossible to directly apply the simplest convolution equation to the data because the backcalculation method requires the count of all AIDS cases over time. Surveillance in Japan has only consistently counted (i) HIV infections without AIDS at the time of diagnosis and (ii) AIDS cases without previous diagnosis of HIV infection. Second, ART has been widespread since 1997 and has continuously improved the prognosis of HIV infection. Explicit incorporation of treatment requires us to account for not only the treatment coverage but also the treatment details (e.g., details of combination therapy), adherence, and many other factors. While there are a number of possible methods to address these issues, including those using CD4 data (e.g., *Van Sighem et al., 2017*) or molecular biomarkers, a simple yet tractable estimation method that rests on epidemiological surveillance data and that can reasonably overcome these problems is called for.

In the present study, the aim was to address the abovementioned issues, estimating the HIV incidence among Japanese nationals, and also to offer statistical estimates of undiagnosed HIV infections and the proportion of diagnosed HIV infections over time.

## MATERIALS & METHODS

### Surveillance data of HIV and AIDS in Japan

The present study investigated the epidemiological surveillance data of HIV and AIDS in Japan, which is publicly reported by the *Committee of AIDS Trends, Ministry of Health, Labor and Welfare, Japan (2018)*, belonging to the Ministry of Health, Labor and Welfare, Japan. Of the reported datasets, our analyses are focused on Japanese nationals because estimation of infection among foreigners requires accounting for human migration, and the decision of migratory behavior (e.g., leaving Japan) is highly dependent on the diagnosis of HIV infection and AIDS. As of the end of 2017, there were 16,663 HIV infections and 7,587 AIDS cases among Japanese nationals (*Committee of AIDS Trends, Ministry of Health, Labor and Welfare, Japan, 2018*). As mentioned, HIV diagnoses reflect HIV-infected individuals who undertook voluntary diagnostic testing before the onset of AIDS. An AIDS case indicates a patient who has never been diagnosed with HIV infection prior to an AIDS diagnosis and who meets the clinical diagnostic criteria: (i) confirmed HIV infection and (ii) the presence of one of 23 indicator diseases representing opportunistic infections or tumors. According to the Infectious Disease Law, HIV and AIDS are classified as a category V notifiable disease, and once diagnosed, physicians must notify the case within 7 days of diagnosis. In the present study, the yearly incidence of HIV infections and AIDS diagnoses from 1985 to 2017 was used. The data are structured by sex and also by the most likely route of transmission (e.g., heterosexual, homosexual or intravenous drug use, based on a physician's interview of patients). The latter information, i.e., the mode of transmission, is discarded because it is believed that a substantial proportion of men having sex with men do not disclose the actual contact and inform physicians that they acquired infection through heterosexual contact (*Inoue et al., 2015*). Thus, a stratified estimation by sex was conducted. Although the magnitude of the epidemic in Japan is relatively small compared with that in Western industrialized countries, the incidence of HIV infection in
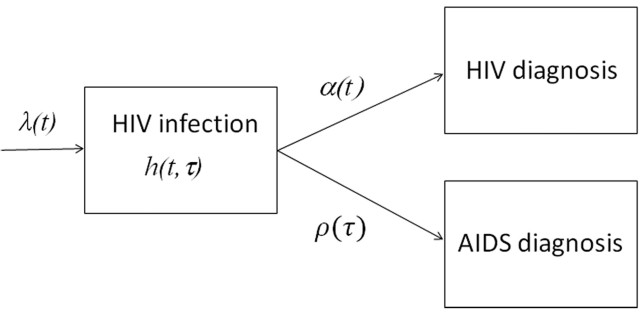

**Figure 1  Data-generating process of HIV infections and AIDS cases in Japan.** New HIV infections occur at rate $\lambda(t)$. While going undiagnosed as $h(t, \tau)$, there would be an increase in the time since infection $\tau$. Diagnosis of HIV takes place at a time-dependent rate $\alpha(t)$, and AIDS illness onset occurs at rate $\rho(\tau)$, which depends on the time since infection. Newly diagnosed HIV infections, and AIDS cases that had not been previously diagnosed with HIV, were notified to the surveillance system.

Japan is believed to have steadily increased over time, especially among men who have sex with men and young adults (*Kihara et al., 2003*; *Nemoto, 2004*).

## Derivation of likelihood using a mathematical model

The proposed statistical model is derived from the following partial differential equation (PDE) model, which is referred to as the McKendrick equation (*Nishiura & Inaba, 2011*; *Ejima, Aihara & Nishiura, 2014*). Figure 1 shows a compartmental diagram of the data-generating process. Once infected with HIV, individuals who are undiagnosed and in the incubation period experience two different hazards, i.e., the force of HIV diagnosis $\alpha(t)$ that depends on calendar time $t$ and the hazard of illness onset $\rho(\tau)$ that depends on the time elapsed since infection $\tau$. Let $h(t, \tau)$ be undiagnosed incubating HIV infections at calendar time $t$ and the time since infection $\tau$ (i.e., undiagnosed HIV infection without AIDS), the dynamics of HIV diagnosis and illness onset are described by

$$\left(\frac{\partial}{\partial t} + \frac{\partial}{\partial s}\right)h(t, s) = -(\alpha(t) + \rho(s))h(t, s), \tag{1}$$

with a boundary condition

$$\lambda(t) := h(t, 0), \tag{2}$$

where $\lambda(t)$ represents the HIV incidence (i.e., the number of new HIV infections) at calendar time $t$. It should be noted that $\rho(\tau)$ yields $f(\tau)$, the probability density function of the incubation period as follows:

$$f(s) = \rho(s)exp\left(-\int_0^s \rho(y)dy\right), \tag{3}$$

for $s > 0$. It is well known that the McKendrick equation can be solved along the characteristic line, i.e.,

$$h(t, s) = \lambda(t - s)exp\left(-\int_{t-s}^t \alpha(x)dx - \int_0^s \rho(y)dy\right), \tag{4}$$

for $t - s > 0$. Eqs. (1) and (4) indicate that the incidence of HIV diagnosis at calendar time $t$, $u(t)$, is written as

$$u(t) = \int_0^t \alpha(t) h(t,s) ds$$
$$= \int_0^t \alpha(t) \lambda(t-s) exp\left(-\int_{t-s}^t \alpha(x) dx - \int_0^s \rho(y) dy\right) ds, \tag{5}$$

and similarly, the incidence of AIDS cases at time $t$, $a(t)$, is

$$a(t) = \int_0^t \rho(s) h(t,s) ds$$
$$= \int_0^t \rho(s) \lambda(t-s) exp\left(-\int_{t-s}^t \alpha(x) dx - \int_0^s \rho(y) dy\right) ds. \tag{6}$$

Equations (5) and (6) read similarly to the so-called extended backcalculation (*Hall et al., 2008*), which is derived from a competing risk model (*Marschner, 1994*; *Cui & Becker, 2000*). The abovementioned process can be used as the generalization.

## Statistical model and estimation

The datasets are reported in a discrete time interval (i.e., year); thus, here I discretized models Eqs. (5) and (6) as

$$u_t = \sum_{s=1}^{t} \lambda_{t-s} \alpha_t \prod_{x=t-s+1}^{t-1} (1-\alpha_x) \prod_{y=1}^{s-1} (1-\rho_y), \tag{7}$$

and

$$a_t = \sum_{s=1}^{t} \lambda_{t-s} \rho_s \prod_{x=t-s+1}^{t-1} (1-\alpha_x) \prod_{y=1}^{s-1} (1-\rho_y). \tag{8}$$

There is no prior notion as to the shape of the epidemic curve (i.e., the frequency of transmission) over time. Thus, the incidence of HIV infection in year $t$, $\lambda_t$, is modeled as a step function:

$$\lambda_t = \begin{cases} \lambda_1 \text{ for } t < 1989, \\ \lambda_2 \text{ for } 1989 \leq t < 1993, \\ \vdots \\ \lambda_9 \text{ for } 2013 \leq t, \end{cases} \tag{9}$$

such that the yearly incidence can be directly dealt with as the parameter. The yearly probability of diagnosis in year $t$, $\alpha_t$, is similarly modeled as

$$\alpha_t = \begin{cases} \alpha_1 \text{ for } t < 1989, \\ \alpha_2 \text{ for } 1989 \leq t < 1993, \\ \vdots \\ \alpha_9 \text{ for } 2013 \leq t. \end{cases} \tag{10}$$

The probability mass function of the incubation period is assumed as known, and in discrete time, this is written as $\rho_s\prod_{y=1}^{s-1}(1-\rho_y)$. As is widely assumed for HIV infection, the incubation period is modeled using the Weibull distribution. Using the property of Weibull distribution with the scale parameter $\eta$ and shape parameter $k$, the discrete Weibull model is connected to the continuous version as

$$\rho_s = 1 - \frac{\exp\left(-\left(\frac{t+1}{\eta}\right)^k\right)}{\exp\left(-\left(\frac{t}{\eta}\right)^k\right)}, \tag{11}$$

and

$$\prod_{y=1}^{t-1}(1-\rho_y) = \exp\left(-\left(\frac{t}{\eta}\right)^k\right). \tag{12}$$

Using the abovementioned model, undiagnosed HIV infections at the end of year $t$ are computed as

$$x_t = \sum_{s=1}^{t}\lambda_{t-s}\prod_{x=t-s+1}^{t-1}(1-\alpha_x)\prod_{y=1}^{s-1}(1-\rho_y). \tag{13}$$

The diagnosed proportion of HIV infections is calculated either as $\sum(a+u)/\sum(x+a+u)$ or $\sum u/\sum(x+u)$, taking the summations over time. The former calculates the proportion of diagnosed HIV-positive individuals out of the cumulative number of HIV-positive individuals. This calculation has the drawback of including patients with AIDS who have already died by the year of calculation. As of 2017, it has been reported that a total of 2,321 cases resulted in death (*Iwamoto et al., 2017*). Alternatively, the latter calculates the fraction of individuals who are HIV positive but have not yet developed AIDS out of the cumulative number of HIV-positive individuals but including undiagnosed individuals, considering that the incubation period in most cases of HIV infection is now considerably extended by ART. The drawback of the latter calculation is that patients with AIDS who have survived and have received ART are excluded; thus, the calculated proportion may not be strictly in line with the target figure in the first goal of the 90-90-90 initiative. Therefore, when estimating the undiagnosed number of HIV infections and the diagnosed proportion at the end of 2017, both calculations are made, and the former is adjusted by subtracting 2,321 AIDS deaths from the cumulative count of AIDS cases.

To quantify the proposed system of equations, we estimate parameters $\lambda_t$ and $\alpha_t$ by means of the maximum likelihood method. Considering that HIV infections are generated as the nonhomogenous Poisson process, the resulting HIV diagnoses and AIDS cases would also follow Poisson distributions. The likelihood function of HIV diagnoses is

$$L_1 = constant \times \prod_{t=1985}^{2017} E(u_t)^{r_t} exp(-E(u_t)), \tag{14}$$

where $r_t$ denotes the reported (observed) number of HIV diagnoses in year $t$ in the surveillance record. Similarly, the likelihood of new AIDS diagnoses is

$$L_2 = constant \times \prod_{t=1985}^{2017} E(a_t)^{w_t} exp(-E(a_t)), \quad (15) \tag{15}$$

where $w_t$ denotes the reported number of new AIDS diagnoses in year $t$. Consequently, the total likelihood $L$ is given by

$$L = L_1 L_2. \tag{16}$$

Maximum likelihood estimates of parameters are obtained by minimizing the negative logarithm of Eq. (16). As mentioned above, the incubation period distribution is assumed as known, and to address the uncertainty, three different estimates are derived from published studies (*Boldson et al., 1988*; *Brookmeyer & Goedert, 1989*; *Munoz & Xu, 1996*). A widely cited estimate by *Brookmeyer & Goedert (1989)* was derived from the study of patients with hemophilia over 20 years of age with $\eta = 11.6$ and $k = 2.5$, resulting in a median incubation period of 10.0 years. *Boldson et al. (1988)* investigated a cohort of AIDS cases in San Francisco with $\eta = 14.3$ and $k = 2.5$, yielding a median incubation period of 12.3 years. The estimate by *Munoz & Xu (1996)* was obtained from the Multicenter AIDS Cohort Study with $\eta = 10.0$ and $k = 1.3$, and the median incubation period is 7.5 years. All three estimates have been used in the present study to address uncertainty with respect to the incubation period. In addition to Eqs. (14) and (15), we have also explored the over-dispersed likelihood function, employing the negative binomial distribution with time-independent dispersion parameter for HIV and AIDS counts, respectively, (*Althaus, 2015*) and compared the Akaike Information Criterion (AIC) against Poisson distributed likelihood, as part of sensitivity analysis.

The 95% confidence interval (CI) of parameters was derived from the profile likelihood. The 95% CI of model estimates (e.g., the number of undiagnosed HIV infections and the proportion diagnosed) was derived using a parametric bootstrap method. In the bootstrapping exercise, model parameters were resampled from a multivariate normal distribution with vectors of mean $\theta$ and standard deviation $\sigma$. The latter vector was derived from the covariance matrix, taking diagonal elements of the inverse Hessian matrix ($\sigma^2 = \text{diag}(H^{-1}(\boldsymbol{\theta}))$). For each set of parameters, the model solution is obtained, and 1,000 times of parameter resampling results in a simulated distribution of model solutions. By taking the 2.5th and 97.5th percentile points of the simulated distribution, the 95% CI is obtained. All statistical data were analyzed using R version 3.1 (Comprehensive R Archive Network) (*R Core Team, 2016*) and JMP version 12.0.1 statistical software (SAS Institute Inc., Cary, NC, USA).

### Ethical considerations

In the present study, the analyzed data are publicly available (*Committee of AIDS Trends, Ministry of Health, Labor and Welfare, Japan, 2018*). As such, the datasets used in our study are deidentified and fully anonymized in advance, and the analysis of publicly available data with no identifying information does not require ethical approval.

## RESULTS

Estimated parameters, i.e., yearly incidence and yearly probability of diagnosis, are shown in Fig. 2. With the assumed median incubation period of 10.0 years, the yearly incidence was the highest from 2005–2008, with an estimated 1,972 (95% CI: 1,829, 2,115) infections per year (Fig. 2A). Subsequently, the incidence began to decline; the yearly estimate in the most recent interval (from 2013–2017) was 1,179 (95% CI: 1,047, 1,293) infections. The yearly probability of diagnosis has monotonously improved over time (Fig. 2B). The estimated diagnosis probability by 1999 was less than 10%, but the latest estimate from 2013–2017 was 15.6% (95% CI: 14.8%, 16.4%). The qualitative patterns of HIV incidence and diagnosis did not vary greatly, even when shorter and longer median incubation periods were used (Figs. 2C and 2D). Figures 2E and 2F show maximum likelihood estimates of the incidence and probability of diagnosis by sex. The incidence in males was the highest from 2005–2008; the latest estimate from 2013–2017 ranged from 1,015 to 1,363 infections per year, assuming a median incubation period from 7.5 to 12.3 years. Similarly, the incidence in females was highest from 1993–1996, ranging from 86 to 97 infections per year; the latest yearly incidence ranged from 31 to 54 infections with a median incubation period of 7.5 to 12.3 years. The yearly probability of diagnosis among males behaved similarly to that of the entire population, but there was no apparent improvement in the frequency of diagnosis among females. In general, female enjoyed higher rate of diagnosis than male. For the entire population with median incubation period at 10.0 years, AIC with the Poisson distributed likelihood was 650.7, while that with negative binomially distributed likelihood was 655.0, indicating that the dataset was not over-dispersed. In fact, dispersion parameters for HIV and AIDS were estimated at greater than 100, indicating that Poisson distribution has sufficiently captured the variation.

Figure 3 shows a comparison between the observed and predicted number of HIV diagnoses and AIDS cases. All three models with different median incubation periods yielded almost identically good fit to the data (Fig. 3A), reflecting mutual compensations between $\lambda$ and $\alpha$ to fit to the data. Even though the number of diagnosed HIV infections and AIDS cases was relatively small for females, the proposed model successfully captured the observed patterns of HIV diagnoses and AIDS cases by sex (Fig. 3B).

Figure 4 shows the estimated undiagnosed number of HIV infections and the estimated proportion of diagnosed HIV-positive individuals over time, among Japanese nationals. Using the median incubation period of 10.0 years (Fig. 4A), undiagnosed HIV infection was estimated to have peaked in 2009 with 7,532 (95% CI: 6,911, 8,152) infections. In the latest time interval, from 2013–2017, it was estimated that 5,363 (95% CI: 4,809, 5,917) infections remained unrecognized. Varying the median incubation period from 7.5 to 12.3 years, the maximum likelihood estimate of undiagnosed HIV infections in the latest time interval ranged from 4,041 to 6,552 infections. These findings indicate that the cumulative number of HIV infections by the end of 2017 was 29,613 (95% CI: 29,059, 30,167) Japanese nationals, using the median incubation period of 10.0 years, and can range from 28,291 to 30,802 individuals.

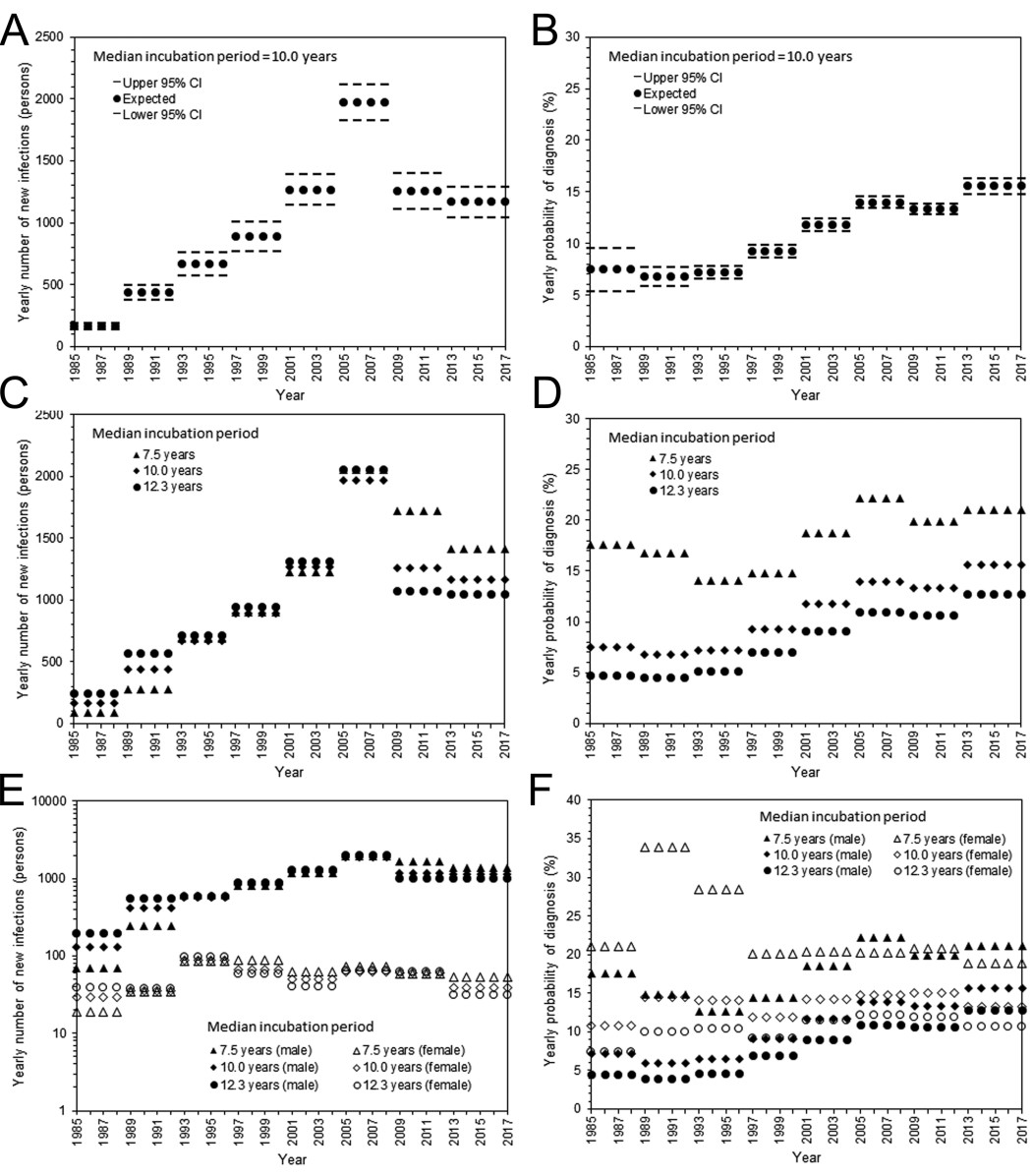

**Figure 2  Estimated HIV incidence and rate of diagnosis in Japan.** (A) The yearly incidence of HIV infection, assuming that the median incubation period is 10.0 years. The step function for every 4 years was used to model the incidence. The 95% confidence intervals were derived from profile likelihood. (B) The yearly rate of diagnosis of HIV infection, assuming that the median incubation period is 10.0 years. (C) Maximum likelihood estimates of the yearly incidence with different median incubation periods: 7.5, 10.0, and 12.3 years. (D) Maximum likelihood estimates of the yearly rate of diagnosis with different median incubation periods: 7.5, 10.0, and 12.3 years. (E) Yearly incidence estimates by sex and different median incubation periods. Maximum likelihood estimates are shown. Note that a common logarithmic scale is used on the vertical axis, to ease comparisons. (F) Yearly rate of diagnosis estimates by sex and different median incubation periods. Maximum likelihood estimates are shown.

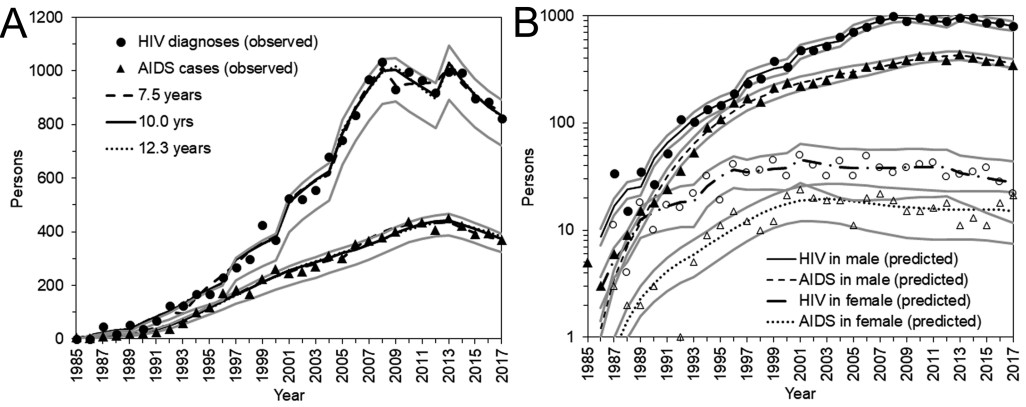

**Figure 3 HIV diagnoses and AIDS cases in Japan, 1985–2017.** (A) Comparisons between observed and predicted yearly number of HIV diagnoses and AIDS cases. Different median incubation periods (i.e., 7.5, 10.0, and 12.3 years) were assumed, but predicted values are mostly overlapped. (B) Comparisons between observed and predicted values by sex. Circles represent the observed number of HIV diagnoses whereas triangles represent that of AIDS cases. Solid marks represent males; empty marks represent females. A common logarithmic scale is used on the vertical axis. In A and B, bold grey lines represent lower and upper 95% confidence intervals with the median incubation period of 10.0 years based on the parametric bootstrap method.

Including and excluding AIDS cases, the estimated proportions of diagnosed HIV infections are shown in Figs. 4C and 4D. Including AIDS cases, the diagnosed proportion was estimated at 81.9% (range 78.7% to 85.7%) using the median incubation period of 10.0 (7.5 to 12.3) years. Excluding AIDS cases, the estimate was 75.7% (range 71.8% to 80.5%). Figures 4E and 4F show the estimated number of undiagnosed HIV infections and the diagnosed proportion by sex, excluding AIDS cases. Estimates of undiagnosed HIV infections among males behaved similarly to the entire population of Japanese nationals, whereas those of females peaked in the year 2001. In the latest time interval (2013–2017), it was estimated that 5,150 infections (range 3,881 to 6,287) in males and 210 infections (range 162 to 255) in females remained unrecognized, using the median incubation period of 10.0 (with the range of 7.5 to 12.3) years. The diagnosed proportion of both males and females increased with time, and females tended to yield higher estimates than males. In the latest time interval from 2013–2017, the diagnosed proportion (excluding AIDS cases) was estimated at 75.3% (range 71.4% to 80.2%) among males and 82.1% (range 79.1% to 85.6%) among females.

Figure 5 shows the undiagnosed number of HIV infections and the proportion of diagnosed infections at the end of 2017. The uncertainty bound was greatest with an assumed median incubation period of 12.3 years, with an estimated 6,552 infections (95% CI: 5,632, 7,471). Figure 5B shows the diagnosed proportion, including and excluding AIDS cases, with 95% confidence intervals. Even when AIDS cases were included, the 2,321 deaths known up to that point were subtracted from AIDS cases in advance of the calculation. Assuming that the median incubation period was 10.0 years, the calculation, inclusive of surviving AIDS cases, yielded 80.3% (95% CI: 78.7%, 82.0%); when excluding AIDS cases, the proportion was 75.7% (95% CI: 73.8%, 77.6%).

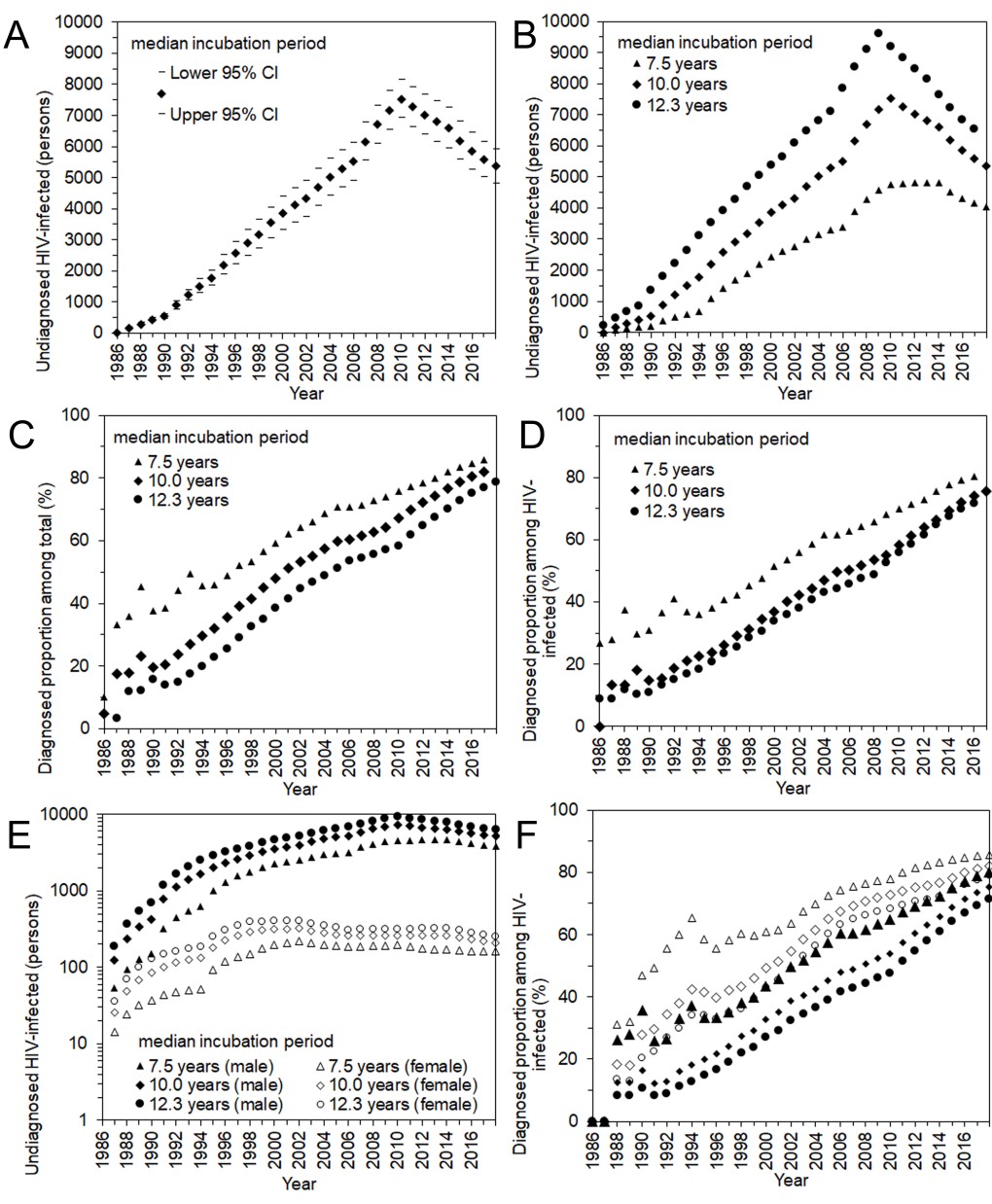

**Figure 4  Undiagnosed number and proportion of HIV infections in Japan, 1986–2017.** (A) Estimates of undiagnosed HIV infections, assuming that the median incubation period is 10.0 years. The 95% confidence intervals were derived from profile likelihood. (B) Maximum likelihood estimates of undiagnosed HIV infections with different median incubation periods: 7.5, 10.0, and 12.3 years. (C) Proportion of diagnosed infections out of the cumulative number of HIV infections, inclusive of AIDS cases. (D) Proportion of diagnosed infections out of the cumulative number of HIV infections, excluding AIDS cases. (E) Maximum likelihood estimates of undiagnosed HIV infections by sex, with different median incubation periods: 7.5, 10.0, and 12.3 years. Note that common logarithmic scale is used on the vertical axis. (D) Proportion of diagnosed infections out of the cumulative number of HIV infections, excluding AIDS cases, by sex.

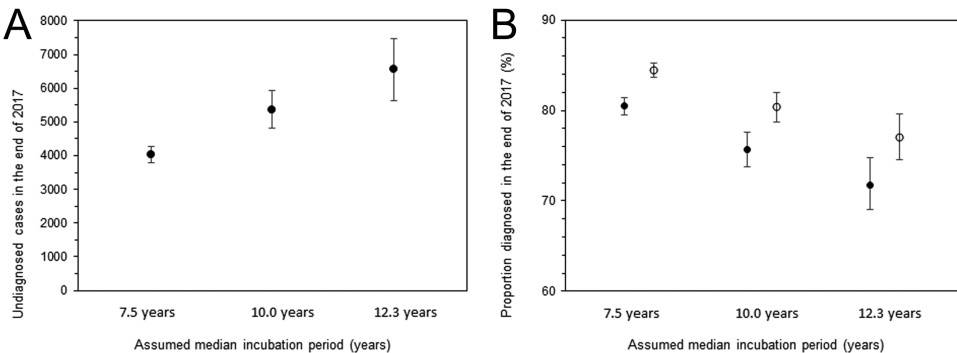

**Figure 5** **Estimated undiagnosed HIV infections and proportion of diagnosed infections at the end of 2017.** (A) Estimates of undiagnosed HIV infections with different incubation periods. Whiskers extend to lower and upper 95% confidence intervals derived using a parametric bootstrapping method. (B) Proportion of diagnosed infections out of the cumulative number of HIV infections, excluding AIDS cases (solid circles) or including AIDS cases but subtracting 2,321 deaths (empty circles). Whiskers extend to lower and upper 95% confidence intervals derived using a parametric bootstrapping method.

## DISCUSSION

The present study estimated the incidence and diagnosed fraction of HIV infections among Japanese nationals, devising an original model that captures the data generating process of HIV and AIDS in the epidemiological surveillance. By the end of 2017, the cumulative number of HIV infections was estimated to be about 30,000 cases, of which 4,000 to 6,000 were considered to have remained undiagnosed. Assuming that the median incubation period was 10.0 years, 80% of infections have ever been diagnosed; accounting for the uncertainty in a median incubation period ranging from 7.5 to 12.3 years, the estimate of the diagnosed proportion can range from 77% to 84%. To the author's knowledge, the present study is the first to offer firm statistical estimates of the incidence and diagnosed proportion of HIV infections based on epidemiological surveillance data in Japan, using an explicit mathematical modeling approach.

There are two take-home messages from the results of this study. First, regardless of whether AIDS cases are included, the proportion of diagnosed HIV infections appears not to have reached 90% among Japanese nationals. Although some estimates exceed 80%, even after subtraction of known deaths owing to AIDS, the findings echo those of a published study that analyzed blood donor data (*Iwamoto et al., 2017*). The published blood donor-based estimate indicated that 85.6% have been diagnosed at the end of 2015, which was prone to sampling bias with substantial potential of both over- and underestimation, and the present study validated that the surveillance-based estimate was slightly below and not too far from the published figure. These findings pose a critical problem in Japan for controlling HIV and AIDS. In the present study, the rate of diagnosis was shown to have improved with time, and the trend was particularly apparent among men, mainly comprising men having sex with men (MSM). The findings of the present study indicate that there would be a certain number of infected individuals who may not have proper access to consultation, testing, and care with privacy protection. To identify the attributes

of such HIV-infected individuals in greater detail, the investigation must be extended to explore heterogeneous patterns, including age-dependence, spatial heterogeneity, and other background characteristics. These are my ongoing research interests.

Second, compared with the peak from 2005–2008, the incidence showed a declining trend. Compared with the estimate in 2005–2008, the upper 95% CIs of the next two time periods (2009–2012 and 2013–2017) were significantly lower than those in the peak period. In fact, a declining trend has also been seen in other datasets, including the incidence of counseling and blood testing at local health centers and the proportion of HIV-positive blood donors over time (*Committee of AIDS Trends, Ministry of Health, Labor and Welfare, Japan, 2018*). The present study results support that these observed declines are partially attributable to actual decreases in the incidence of HIV infection in Japan. The underlying mechanisms of such decreases have yet to be explored using a mathematical model, perhaps requiring modeling of the saturated effect (*Heesterbeek & Metz, 1993*) together with statistical estimates of the effective reproduction number (*Kretzschmar et al., 2013*). The success of Japan's controlling HIV transmission among the core population, i.e., MSM, reflecting the set-up of gay community centers and scale-up of gay non-governmental organizations' activities (*Sherriff et al., 2017*), could potentially be objectively demonstrated in such an analysis. In addition, it must be remembered that the yearly incidence still remains above 1,000 infections; moreover, such a declining trend is not evident among females, although the rate of diagnosis in female has steadily been higher than that in male. Higher frequency of diagnosis among female than male might reflect better awareness of the risk that results from heterosexual transmission (e.g., through foreign partner's diagnosis).

Although the present study was motivated by the need for quantifying the care cascade in Japan, in accordance with the goals of 90-90-90, a few technical issues must be noted to interpret the estimates and apply the present results to the evaluation. First, Japanese estimates of the latter two goals of the 90-90-90 initiative, i.e., access to ART and virus suppression, rest on questionnaire surveys conducted in the prefectures, which do not distinguish between infected individuals who are Japanese nationals and those who are not (*Iwamoto et al., 2017*). Thus, our estimates of the diagnosed proportion of HIV infections among Japanese nationals alone cannot immediately be compared with subsequent existing proportions as if they were sampled from the same population. Whereas estimation of the HIV incidence among non-Japanese nationals is an ongoing research subject, it is frequently the case that infection with HIV or illness onset of AIDS acts as a trigger for foreigners to leave the country; therefore, incorporation of their involvement in the transmission dynamics of Japan requires that very careful attention be paid to migration events (*Matsuyama et al., 2018*; *Sakamoto et al., 2018*; *Yuan & Nishiura, 2018*), and ideally, that information is supported by individual-based data. Second, the clinical definition of AIDS in Japan depends on indicator diseases, imposing a certain extent of uncertainty in diagnosis. For instance, Japan has a number of designated AIDS Core Hospitals, and HIV diagnoses in those institutes involve screening of common opportunistic infections upon diagnosis of HIV infection, which sensitively leads to the diagnosis of AIDS. Compared with HIV-infected individuals diagnosed at local health centers, the frequency of AIDS

diagnosis may be higher in the designated hospitals, calling for the validation of estimates using other methods. Using other datasets including biomarkers or CD4 count data can act as another potential work to be built on this study. Third, in the present study, we struggled with subtraction of AIDS deaths from the calculation of the diagnosed proportion of infections; this problem essentially stems from the absence of a case registration system in Japan. Once diagnosed, infected individuals are never longitudinally monitored by the government, considerably complicating prevalence estimation. With a registration system of HIV-infected individuals, statistical monitoring of the second and third goals of 90-90-90 can be achieved in real time and in a very reasonable manner.

Five technical limitations must be noted. First, the present study did not account for uncertainties other than variations in length of the incubation period. There has been a concern that the incubation period has probably shortened over time (*Nakamura et al., 2011*), but I did not have substantial data to support this issue. Second, the natural history of HIV infection has yet to be explored in-depth; an explicit proportion of HIV individuals who never develop AIDS over the course of infection is missing. Third, other than sex, the present study accepted homogeneity in the natural course and diagnosis of infection. Our future studies will address several heterogeneities. Fourth, estimates rested on yearly data, and the precision was limited (e.g., with use of the step function for every 4 years). The use of smoothing with nonparametric back-projection is another of our ongoing studies (*Becker, 1997*). Fifth, the present study focused on the incidence estimation, and more explicit modeling of the transmission dynamics, including those highlighting the role of MSM (*Yamamoto, Ejima & Nishiura, 2018*), is the subject for future studies.

Despite these limitations, the present study successfully estimated the incidence of HIV infections, undiagnosed number of infections, and the proportion diagnosed in real time, using limited but readily available epidemiological surveillance data. Improved estimates using age and geographical data, as well as estimates based on other methods, are to follow, which will boost studies of epidemiological estimation in this area in Japan.

## CONCLUSIONS

In the present study, a statistical modeling method was developed for the estimation of HIV incidence in Japan and estimates made of the undiagnosed number of HIV infections and the proportion of diagnosed HIV infections over time. Using the McKendrick equation, a general convolution-like equation was derived, allowing for joint estimation of the HIV incidence and time-dependent rate of diagnosis. By the end of 2017, the cumulative number of HIV infections was estimated to be about 30,000, and about 80% of infections have ever been diagnosed. Accounting for the uncertainty in the median incubation period ranging from 7.5 to 12.3 years, estimates of the diagnosed proportion of HIV infections can range from 77% to 84%. The proportion of diagnosed HIV infections appears not to have reached 90% among Japanese nationals.

## ACKNOWLEDGEMENTS

We thank Analisa Avila, ELS, of Edanz Group for editing a draft of this manuscript.

### Funding

The present study was primarily supported by the Health and Labour Sciences Research Grant (H26-AIDS-YoungInvestigator-004). The author also received financial support from Health and Labour Sciences Research Grant (H28-AIDS-General-001), the Japan Agency for Medical Research and Development (grant number JP18fk0108050), Japanese Society for the Promotion of Science (JSPS) Grant-in-Aid for Scientific Research (grant numbers 16KT0130 and 17H04701), and Japan Science and Technology Agency (JST) Core Research for Evolutional Science and Technology program (JPMJCR1413). The funders had no role in study design, data collection and analysis, decision to publish, or preparation of the manuscript.

### Grant Disclosures

The following grant information was disclosed by the author:
Health and Labour Sciences Research Grant: H26-AIDS-YoungInvestigator-004.
Health and Labour Sciences Research Grant: H28-AIDS-General-001.
Japan Agency for Medical Research and Development: JP18fk0108050.
Japanese Society for the Promotion of Science (JSPS) Grant-in-Aid for Scientific Research: 16KT0130, 17H04701.
Japan Science and Technology Agency (JST) Core Research for Evolutional Science and Technology program: JPMJCR1413.

### Competing Interests

Hiroshi Nishiura is an Academic Editor for PeerJ and has no competing interests.

### Author Contributions

- Hiroshi Nishiura conceived and designed the experiments, performed the experiments, analyzed the data, contributed reagents/materials/analysis tools, prepared figures and/or tables, authored or reviewed drafts of the paper, approved the final draft.

### Data Availability

The raw data with the website URL is referenced in the manuscript, and any readers can download the data from there.

http://api-net.jfap.or.jp/status/2016/16nenpo/hyo_03.pdf.

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
