# Peer review of "Estimating the incidence and diagnosed proportion of HIV infections in Japan: a statistical modeling study"

_PeerJ, doi:10.7717/peerj.6275_

## Round 0.1 · original submission · Minor Revisions

While both reviewers highlighted the usefulness and applicability of the presented method to estimate HIV incidence, they also raised some comments that could be quickly addressed in a revised version of your manuscript. Please take particular attention to the comment by Reviewer 1 regarding uncertainty and the relatively small confidence intervals in your estimates.

·

Basic reporting

English: good
References/context: good
Structure: good
Data: publicly available (not clear if location is provided)
Figures/tables quality: good
Self-contained: yes

Experimental design

Scope of the journal: adapted
Research question: important and interesting, with real-world implications
Methods: rigorous and adapted to the research question, well described

Validity of the findings

Statistical analysis: sound and robust
Conclusions supported by analysis: yes

Additional comments

Prof. Nishiura adapts a method based on McKendrick and Morison's approach to estimate the incubation period during the 1919 flu epidemic to address a problem related to specificities of the surveillance of HIV in Japan. Indeed, surveillance of HIV in this country relies upon 1) reports of new HIV diagnosis among people who undertook voluntary testing and 2) reports of new AIDS cases among people that were not diagnosed as HIV-infected. This data does not allow the direct estimation of HIV incidence using traditional back-calculation methods, as the incidence of all AIDS cases is not available.

The proposed method method is generative and closely matches the mechanisms of HIV surveillance and evolution towards AIDS (and this could be more emphasized in the paper). It allows the estimation of past HIV incidence in a convincing way.

My only concern about this work is the handling of uncertainty. It is surprising to me that the data allows e.g. the estimation of the cumulative number of HIV infection with such a narrow confidence interval (29,059-30,167). Seeing Fig. 3, this may be related to overfitting. In that case, using a negative binomial distribution instead of a Poisson might have been preferable. The author should provide a sensitivity analysis using a negative binomial distribution, or support his use of the Poisson distribution.

+++++
Minor comments

l121: while the model is well described both in equations and in Fig. 1, the text is a bit confusing as to
which population does compartment h(t,tau) represent precisely. Line 121, it is written that it is "individuals who are either undiagnosed or in the incubation period" and it seems to me that it should be "individuals who are both undiagnosed *and* in the incubation period", as either of these events would remove an individual from the h compartment. The terms referring to HIV diagnosis and AIDS diagnosis should also be more coherent, as in the current state of the manuscript the authors sometimes use "incubating" or just "diagnosed".

l130: "rho(tau)" should be "f(tau)"

l135: I don't understand why the upper boundary of the second integral is x, shouldn't it be s? (also in following equations)

l136: It would maybe make the equations easier to read to use h(t,s) instead of the full expression in eq. 5 and 6 as h(t,s) is defined in eq. 4.

l150: there is no justification as to why a step function has to be used, is that related to data structure?

l160: very minor, it is not clear in the notation that k is an exposant

Fig2F: this plot is very confusing, the author should consider using a different coding, or maybe remove part of the information displayed.

l237: the predicted values obtained with 3 different median incubation periods are very close, although they lead to marked differences in the estimated incidence of HIV (Fig 2C). Could the author provide an explanation about this? Also, I would have liked to see confidence intervals on Fig. 3A and 3B, although I understand that it is not convenient with only one figure.

l288: the comparison of the results to Iwamoto (2017) should be a bit more detailed and discussed.

Reviewer 2 ·

Basic reporting

No main comments, otherwise please see general comments below.

Experimental design

No comment

Validity of the findings

No comment

Additional comments

This is a timely, useful and practical analysis of relevance for characterizing the HIV epidemic in Japan and for informing HIV policy and programming in this country. The analysis is thoughtfully and finely conducted and captures the essential features to address the research questions, and nicely using a parsimonious model. The complexity of the analytical methods is appropriate for the research questions and nature of available data.
I have few remarks in the spirit of enhancing the article’s contribution to the literature and its impact. These can be addressed through a minor revision.
1) Introduction Line 69: I think it would be useful here to indicate the limitation of potential under-sampling in blood donor data. For example, HIV testing for blood donors in the US substantially underestimates actual HIV prevalence in the population. Accordingly, the Iwamoto et al 2017 analysis could have underestimated the prevalence of infection, and thus overestimated the fraction diagnosed.
2) Introduction Line 86: In relation to estimation methods using HIV surveillance data, there are currently different developed methods, the use of which depends on the nature of available data. For example, the ECDC HIV Modelling Tool provides an example of available methods that are useful with availability of CD4 data (see van Sighem et al, Euro Surveill. 2017). Meanwhile, the method discussed in the present paper is useful and fitting in consideration of Japan’s data. I suggest more discussion of this issue, such as along the lines discussed here.
3) Methods Line 121: The statement “either undiagnosed or in incubation period” seems confusing to me. I think it is just simpler to state it as “undiagnosed HIV infected persons who have not developed AIDS”, or something along this line.
4) Results Line 235: It is interesting that there was no improvement in the frequency of diagnosis among females. This is worth discussing in the Discussion section and providing potential explanation(s). For example, could this relate to how females are being typically diagnosed, say through routine testing rather than VCT services?
5) Results Line 264: The higher diagnosis rate among females compared to males is worth discussing/explaining in the Discussion section (actually it is paradoxical and counter-intuitive). Why is this the case, even though the focus of VCT services in Japan is to target MSM, rather than women/heterosexual populations? Could this relate to how females are being actually diagnosed (say following the diagnosis of their spouse/partner)?
6) Discussion: A gap in the Discussion section is linking the results to the HIV epidemic among MSM in Japan, which is the core of the epidemic. Much of HIV incidence is believed to occur among MSM, but the paper hardly discusses this issue. What is and how did HIV prevalence among MSM change over time in Japan? Are there other and contextual data of the epidemic among MSM that can explain the results of this paper?
7) Discussion: A related point, which is a limitation of the present study, is that the modeling method is on the descriptive indirect side, and does not directly model HIV transmission, such as among MSM, to generate such results. It is worth suggesting in the Discussion section that future work should build on this work to model HIV transmission in the population, with focus on the role of MSM.
8) Discussion: Another potential work that could build on this study is to attempt to use other data, such as CD4 count data among diagnosed infections, to generate similar results and examine the differences if any. Such application can also be useful to generate inferences about HIV cascade, which is critical to inform HIV response. I suggest discussing this point.
9) Discussion Line 304: The declines in HIV incidence is probably best explained by examining HIV transmission dynamics among MSM. I am aware of an earlier work to model the HIV epidemic among MSM in Japan (regrettably unpublished), that reached similar conclusions about the trend in HIV incidence, though through a very different approach. This is also supported by the scale-up of NGOs working with MSM in Japan. The authors may want to discuss this issue in some detail, as it provides important epidemiological context.
10) Please check the citations and references. For example, I do not seem to find the bibliography entry for the citation of Hall et al 2008.
11) Typo in Results Line 224: probability of diagnosis “has” monotonously improved…

---

## Round 0.2 · Minor Revisions

Thank you very much for responding to the reviewers' comments in the revised version of your manuscript.

I agree with the final comment by reviewer 1 that the obtained difference in AIC between the Poisson and negative binomial model is odd. The negative binomial model should fit at least as good as the Poisson model, so the difference in AIC should not be more than 2 (for the single additional parameter). The large difference between the AIC's that you report could be a result of the negative binomial model failing to find a global optima. If that is the case, I suggest you either perform model fits using a different starting value for the dispersion parameter (e.g., a very large value), or you mention that fitting the negative-binomial model fails to converge to a global optima.

I also agree with reviewer 1 that citing the papers by Riou et al. (2018) and Althaus (2015) as a reference for over-dispersed case counts is not appropriate. An alternative reference for the use of negative binomial distributions in epidemiology would be Lloyd-Smith (PLoS One. 2007 Feb 14;2(2):e180).

·

Basic reporting

No comment.

Experimental design

No comment.

Validity of the findings

No comment.

Additional comments

Prof. Nishiura modified his paper according to the reviewers' recommendations. The new version is mostly satisfying, only one point remains questionable in my opinion.

The author compared his findings based on a Poisson distribution with an alternative approach using a negative-binomial distribution (line 249 of the new manuscript). I fail to understand how the AIC could be 847.8 using a NB distribution versus 650.7 with a Poisson distribution. If the Poisson is indeed more adapted to the question, shouldn't the overdispersion parameter of the NB distribution go to infinity so that the NB reduces to Poisson, and the AIC a bit higher because of the additional parameter? Could this very large difference between Poisson and NB be caused by a local optima found by the maximum likelihood algorithm? Adding a short clarification of this point would be enough to accept the manuscript for publication.

Other minor comments:
- citing my paper as a reference for approaches using negative binomial distributions is a bit excessive. Considering that I am reviewing this manuscript, I would be more confortable if this citation was replaced by a more general article on overdispersion for count data or simply removed.

Reviewer 2 ·

Basic reporting

The revision addressed my comments.

Experimental design

The revision addressed my comments.

Validity of the findings

The revision addressed my comments.

Additional comments

The revision addressed my comments.

---

## Round 0.3 · accepted · Accept

Thank you very much for responding to the last issue regarding the negative binomial distribution.